# Design and Parametric Analysis of a Wide-Angle and Polarization Insensitive Ultra-Broadband Metamaterial Absorber for Visible Optical Wavelength Applications

**DOI:** 10.3390/nano12234253

**Published:** 2022-11-29

**Authors:** Md Zikrul Bari Chowdhury, Mohammad Tariqul Islam, Ahasanul Hoque, Ahmed S. Alshammari, Ahmed Alzamil, Haitham Alsaif, Badr M. Alshammari, Ismail Hossain, Md Samsuzzaman

**Affiliations:** 1Department of Electrical, Electronic and Systems Engineering, Faculty of Engineering and Built Environment, Universiti Kebangsaan Malaysia, Bangi 43600, Selangor, Malaysia; 2Department of Electrical Engineering, College of Engineering, University of Ha’il, Ha’il 81481, Saudi Arabia; 3Space Science Center (ANGKASA), Universiti Kebangsaan Malaysia, Bangi 43600, Selangor, Malaysia; 4Department of Computer and Communication Engineering, Faculty of Computer Science and Engineering, Patuakhali Science and Technology University, Patuakhali 8602, Bangladesh; 5Department of General Educational Development (GED), Faculty of Science and Information Technology (FSIT), Daffodil International University, Dhaka 1341, Bangladesh

**Keywords:** metamaterial absorber, bendable, oblique incident stable, visible optical wavelength, polarization insensitive

## Abstract

Researchers are trying to work out how to make a broadband response metamaterial absorber (MMA). Electromagnetic (EM) waves that can pass through the atmosphere and reach the ground are most commonly used in the visible frequency range. In addition, they are used to detect faults, inspect tapped live-powered components, electrical failures, and thermal leaking hot spots. This research provides a numerical analysis of a compact split ring resonator (SRR) and circular ring resonator (CRR) based metamaterial absorber (MMA) using a three-layer substrate material configuration for wideband visible optical wavelength applications. The proposed metamaterial absorber has an overall unit cell size of 800 nm × 800 nm × 175 nm in both TE and TM mode simulations and it achieved above 80% absorbance in the visible spectrums from 450 nm to 650 nm wavelength. The proposed MA performed a maximum absorptivity of 99.99% at 557 nm. In addition, the steady absorption property has a broad range of oblique incidence angle stability. The polarization conversion ratio (PCR) is evaluated to ensure that the MMA is perfect. Both TM and TE modes can observe polarization insensitivity and wide-angle incidence angle stability with 18° bending effects. Moreover, a structural study using electric and magnetic fields was carried out to better understand the MMA’s absorption properties. The observable novelty of the proposed metamaterial is compact in size compared with reference paper, and it achieves an average absorbance of 91.82% for visible optical wavelength. The proposed MMA also has bendable properties. The proposed MMA validation has been done by two numerical simulation software. The MMA has diverse applications, such as color image, wide-angle stability, substantial absorption, absolute invisible layers, thermal imaging, and magnetic resonance imaging (MRI) applications.

## 1. Introduction

The advanced electromagnetic properties of metamaterials have been investigated as part of metamaterial research. As early as 1968, V. Veselago explained materials with negative permittivity and permeability are not generally matched by electrodynamics [1]. Metamaterials are artificial composites that unexpectedly interact with light and sound waves compared to natural materials, which has inspired many researchers to design a unique contribution path in practically all fields [2]. 

The physical structure of MM determines its properties, not its chemical content. Since it may be used in a variety of applications, such as absorber [3], polarization converter [4], filter [5], imaging [6], antenna [7], invisible clock [8], sensor [9], waveguides [10], etc. The perfect metamaterial absorber was first developed by Landy [11], which started an innovative study area for its unique use in many applications, such as photo-detecting [12], energy harvesting [13], thermal emitter [14], and solar cell [15], etc. Researchers are now pursuing research on the metamaterial perfect absorber design for microwave frequency, visible, infrared, and THz applications and developing low-loss devices by structural optimization [16]. Furthermore, the researcher was attracted to solar energy collecting by the MPA of total visible light, which is 50% of the total visible light [17,18]. The asymmetrical absorption property can be achieved by using appropriate magnetic and electric resonance engineering. Most researchers have demonstrated various types of metamaterial perfect absorber designs with lower absorption features, such as limited frequency range and lower absorption label [19]. The complete visible optical wavelength range has only been studied in a small number of research concerning the frequency range from 400–790 THz in the visible wavelength range of 380–750 nm. The unique absorption rate at various polarization incident angles required insensitive polarization features for absorber design [20]. Furthermore, It is also essential to have a broad range of incident angle stability to achieve the optimal absorption rate. Zhengqi Liu et al. [21] designed a metamaterial absorber with a maximum absorptivity of 92% by utilizing Au and SiO_2_ metal. The visible wavelength range of 370–880 nm has absorption values over 83%. A four-layer optical wavelength absorber based on Cu, Si_3_N_4_, and Si materials is constructed for 400–700 nm in work [22], with a peak absorption of 97% and a low absorption of approximately 80%. S. Mahmud reported in work [23] that a three-layered metamaterial absorber for a visible range with an absorption bandwidth ranging from 389.34–697.19 nm and absorptivity of 99% was achieved with absorption bandwidths exceeding 91.24%. The absorption level is increased to 70% when the polarization is insensitive, and an incident angle of 60 degrees stability is observed. Work [24] presents a small-size absorber based on Ag and SiO_2_ with a reduced absorption level and absorption bandwidth, but the maximum absorptivity value is enhanced to 98% compared to the previous work, where the wavelength range was 300–700 nm. However, the oblique incidence angle stability and polarization insensitivity are absent. The stability of the incidence angle of an Au and Si-based absorber is raised to 65 degrees in the article [25]. However, absorption drops to 80%. In work [26], the author proposed a Ni and Si base absorber for wavelengths ranging from 400 to 700 nm, with more than 90% absorption intensity. After reviewing the previous studies, it is evident that the oblique incidence angle stable MMA and polarization insensitivity with a maximum absorptivity level throughout the entire wavelength range from 380–750 nm is highly required for visible optical MMA applications.

This research presents a simple metal-dielectric-metal structured MMA for the optical spectrum from 400 to 750 nm. The PCR is evaluated to ensure that the MMA is perfect. The absorbance characteristics of both the TM and TE modes have been investigated at various polarization angles and oblique incidence angles. The results demonstrate that the absorbance properties of both TM and TE modes are unique. The proposed MMA achieved a maximum absorptivity of 99.99% at the working spectrum with more than 91.82% average absorbance levels. The paper is arranged into different sections for the convenience of presentation. In Section 1, the introduction is given. In Section 2, both the design procedure and the structure are discussed. Section 3 provides result analysis and discussion. Finally, the conclusion of the work can be found in Section 5.

## 2. Design and Simulation Setup

### 2.1. Materials Choice

The proposed unit cell has a split ring resonator (SRR) and circular ring resonator (CRR) on top of dielectric material. The backside layer is fully blocked with a dielectric material. Tungsten serves as the metal, and silicon dioxide (SiO_2_) is chosen for the dielectric. The scattering condition, band structure, and optical characteristic of silicon dioxide (SiO_2_) and tungsten (W) are detailed in this article by G. Ghosh [27,28]. Tungsten is significantly more stable for applications involving a wide frequency band in the optical range, when compared to other commonly used metals in MA. The superior impedance matching characteristics that tungsten possesses in the optical range are the primary reason for selecting it as the material to be used in the resonator layer, which reduces transmission and reflection. The loss-free feature silicon dioxide (SiO_2_) is chosen for the dielectric layer. Tungsten shows flexible absorptivity inside the optical region, and resonance properties help to design impedance matches. Optical wavelengths are essentially lossless; therefore, they ignore absorption. As a result, this geometrical framework is crucial for entrapping the resonance wavelength. Metamaterials, such as SRR and CRR, use an organic or inorganic layer to modify absorption by modifying the resonator structure. The excitation coefficient and the refractive index of a typical sample of tungsten are 2.916877 and 3.63739, respectively, when determined at 557 nm; it is also known as the mid optical window band, and there are no excitation coefficients when viewed from the opposite side of silicon dioxide (SiO_2_); the refractive index is 1.45704 at 557 nm [29]. 

### 2.2. Design of the Unit Cell Structure

The proposed design depicts a metal-dielectric-metal MMA structure with a patch and ground of tungsten (W) and a silicon dioxide (SiO_2_) dielectric substrate. The width and length of the unit cell that has a square shape are both defined to be 800 nm. The outer split ring resonator (SRR) is defined as W_1_ = 680 nm and L_1_ = 306 nm, and the inner split ring resonator (SRR) is defined as W_2_ = 480 nm and L_2_ = 205 nm. The parameters are correspondingly specified as H_1_ = 100 nm, H_2_ = 60 nm, and H_3_ = 15 nm. The CRR’s radius is R_1_ = 140 nm and R_2_ = 160 nm. The surface area, (δ) = (2/2fR0)1/2, must be larger than the thickness of the metallic layer to achieve near-zero transmission in an optical area. Figure 1 shows the unit cell’s front, side, back, and perspective views. Tungsten (W) and silicon dioxide (SiO_2_) can be identified by their respective light blue and red colors. Table 1 provides a summary of the overall design parameters. An open add space on the z-axis unit cell boundary conditions have been imposed, as well as the x-axis and y-axis for simulation. The refractive index and excitation coefficient of tungsten and Silicon dioxide (SiO_2_) in the optical are illustrated in Figure 2.

### 2.3. Numerical Analysis Setup

A proper and correct simulation setup is required to obtain the necessary absorbance from the dispersing value. CST Microwave Studio 2019 used terahertz mesh in a frequency-domain solver to perform numerical simulations. The proposed MMA’s boundary conditions and array structure are depicted in Figure 3a, and Figure 3c reveals the proposed MMA’s tetrahedral mesh properties. For transverse electromagnetic (TEM) propagation mode, an installation of a port point is created using both the positive and negative *z*-axis. The impedance matching process between the incident wave and the meta surface takes along the positive *z*-axis. In addition, the *x* and *y* axes are constrained, whereas the positive *z*-axis is regarded as an open add space for the TEM propagation mode. Moreover, *y*-axis and the *x*- axis boundaries are specified as perfect magnetic conductors (PMCs) and electric conductors (PECs). The magnetic field is asymmetrical in the PMC and symmetrical in the PEC, but the electric field is asymmetrical in the PEC and symmetrical in the PMC. Since the z-axis is considered to be a free space with a perfectly matched layer (PML) in the opposite direction, scattering is reduced along with the operational wavelength. The configuration has all three axes operating in opposite directions from one another. For this simulation, we assume that a planar, wide-spectrum wave, and linearly polarized will reach the top of the proposed absorber. The proposed design has been protected when using a perfect electric and magnetic conductor, an electromagnetic response diagram of a crossed rectangle array that is polarization insensitive. As shown in Figure 3b, the optical responses for both the TM and TE propagation modes are identical for these symmetrical 3 × 3 array arrangements. The *x* and *y* axes establish the unit cell’s TE and TM mode boundary conditions. When a higher mesh order is applied, it is possible for the simulation’s results to be more accurate. All the data has been obtained using frequency domain analysis. In addition, a time-domain analysis may be performed on this specific type of absorber simulation. A tetrahedral mesh was combined with an adaptive mesh generation technique to perform the simulation. The substrate’s dielectric properties are estimated to have a free space propagation impedance of approximately 377 Ω. The quality of this tetrahedral mesh analysis goes somewhere between 0.0300454 and 0.997996 and the lowest and highest corner values are 5.9427 nm and 176.343 nm, respectively. In order to achieve the most accurate results from the simulation, a mesh with a total of 27,821 tetrahedrons and an average quality of 0.757251 was used. Regarding the convergence of the simulation process that CST MWS performed, we did not have any problems.

### 2.4. Design Evaluation

The proposed design has a split ring resonator (SRR) and a circular ring resonator (CRR) which is shown in Figure 4a. Figure 4b shows the different layers of absorbance properties, whereas the SiO_2_ dielectric layer shows there is no absorbance. As a result, all incident waves are transferred through the dielectric substance. Although the addition of a patch on metal results in approximately 63.7% absorbance and the highest incident wave continues to pass through the material. The patch and middle layer show 99.99% of maximum absorbance, but the average absorbance for the patch and middle layer is lower. An average absorbance level of 90.82% is attained throughout the working wavelength when the tungsten (W) is in both the top and bottom planes when a dielectric substrate layer (SiO_2_) is mounted between the top and bottom planes. It’s inserted to block the transmission of the incident wave. Moreover, as seen in the design evolution, the geometrical shape of the patch is essential to achieve greater and broader absorbance levels of the suggested MMA. Figure 4c depicts the absorbance properties of different suggested MMA design progressions. The study delves into the vital role that the SRR and CRR of the unit cell perform. In the first step, This model has a peak absorbance of 95.37% at a visible spectrum of 557 nm with no other shape in the center of the unit cell. This configuration has an average bandwidth of 47 nm between 450 and 650 nm. In the meantime, the SRR is stuffed with another square ring resonator.

Furthermore, the average absorbance between 450 nm and 650 nm is 80% due to the intervention of this square ring resonator, the peak absorbance is 96.02%, and the bandwidth is expanded from 47 to 82 nm. The peak absorbance was increased from 96.2% to 97.4% after the two-square ring resonator was split in the third step. After loading a circular ring resonator, the bandwidth was expanded from 75 nm to 119 nm, and peak absorbance was raised to 99.98% at 555 nm. In these scenarios, its average absorbance is significantly lower than expected. Finally, in the last step, the structure reaches maximum absorbance of 99.99% at a wavelength of 557 nm when two metallic splits are loaded with a circular ring resonator. In addition, the suggested model has an average absorbance of 97.82%, along with sensitivity to polarization and properties relating to the angle of incidence.

## 3. Absorption Characteristics

Tungsten has a very excellent demonstration to impedance matching in open space, and it does not have any surface plasmon resonance at wavelengths of the optical region. Tungsten is used both for the resonator and back layers in the proposed MMA absorber. This absorber must function as a high-quality potential super absorber since Z_0_ is slightly less than Z(ω). The electromagnetic waves in an incident are handled by a symmetrical design. Metal plane wave occurrences in the rear layer indicate lower transmissions, and front wave resonator and dielectric layer maintenance need further layers. Silicon dioxide (SiO_2_) effectively decreases the wavelength distance because of its common thickness properties. Impedance matching with the open space optical range may also be determined through the characteristics of the basic losses of the front layer as well as the resonance of the dielectric layers. This uniqueness is achieved by implanting the matching wave absorber, which has excellent performance and efficiency. Moreover, these mysteries will reveal excellent absorbance if the equally structured split ring resonator is employed. The wave from the first stage appears in the resonator as a metal screen reflects the wave in a back layer. A metal resonator and a silicon dioxide (SiO_2_) dielectric help to compensate for the front layer, while a metal plane forms the bottom (tungsten). Figure 5 illustrates the reflectance, transmission, and absorbance plot, and Figure 6 shows the absorbance characteristics comparison between TE, TM, and TEM modes. The suggested MMA has a maximum absorbance of 99.99% at 557 nm and an average absorbance of 91.82% from 450 to 650 nm at TM and TE mode.

On the other hand, in TEM mode, there is a wavelength range of 450 to 650 nanometers with an average absorbance of 91.31%. Metamaterial absorbers have a massive decrease in scattered light from ultraviolet to near-infrared. Research in near-infrared using reflectance imaging and transillumination with optical consistency has shown that this area of the spectrum is well suited for observing materials with a high absorbance polarization. Reflectance measurements may benefit greatly from optimizing the contrast of polarization due to the transparency of absorbers constructed from growing metamaterials. This factor also affects transillumination lesions for TE mode electromagnetic waves as opposed to TM mode. In addition, the absorber has a substantially higher level of light scattering than is usual for materials because it is highly anisotropic and has a unique dependence on light distribution across multiple wavelengths. This research aimed to examine absorbers’ attenuation and absorbance of optical-range electromagnetic waves. The attenuation was discovered to rise dramatically with wavelength, which was not always in line with the expected increase in EM wave absorbance. As a result, PCR shifts between TE and TM modes at a lower wavelength. The absorbance property A (ω) has been determined with the help of Equation (1) [30]
(1)$$A\;(\omega )=1-{\left|S11\left(\omega \right)\right|}^{2}-{\left|S21\left(\omega \right)\right|}^{2}=1-R-T$$
where R and T are the reflectance and transmittance, respectively, S11 is the reflection coefficient, and S21 is the transmission coefficient.
(2)$$Z=\sqrt{\frac{{\left(1+S11\right)}^{2}-S21^{2}}{{\left(1-S11\right)}^{2}-S21^{2}}}=\sqrt{\frac{\frac{\mu }{\varepsilon }}{Z_{0}}}=\sqrt{\frac{\mu_{r}}{\varepsilon_{r}}}$$
where ε = $\varepsilon_{r}\varepsilon$ and μ = $\mu_{r}\mu_{0}$ are the permittivity and permeability of MMA. The $\varepsilon_{r}$ and $\mu_{r}$ are the relative permittivity and permeability of the medium, respectively, and $\varepsilon_{0}\;\mathrm{and}\;\mu_{0}$ are the permittivity and permeability of free space, respectively.

So, free space impedance, $Z_{0}=\sqrt{\frac{\mu_{0}}{\varepsilon_{0}}}$. The reflectance (R) of TM and TE mode can be computed with the help of Equations (3) and (4) [31]
(3)$$R_{TE}={\left|r_{TE}\right|}^{2}={\left|\frac{\mu_{r}\cos\theta -\sqrt{n^{2}-\sin\theta }}{\mu_{r}\cos\theta +\sqrt{n^{2}-\sin\theta }}\right|}^{2}$$
(4)$$R_{TE}={\left|r_{TE}\right|}^{2}={\left|\frac{\varepsilon_{r}\cos\theta -\sqrt{n^{2}-\sin\theta }}{\varepsilon_{r}\cos\theta +\sqrt{n^{2}-\sin\theta }}\right|}^{2}$$
where *n* is the refractive index, and *θ* is the incident angle of the wave for the incident.

As a result, Equations (3) and (4) become
(5)$$R_{TE,TM}={\left|\frac{Z-Z_{0}}{Z+Z_{0}}\right|}^{2}={\left|\frac{\sqrt{\mu_{r}}-\sqrt{\varepsilon_{r}}}{\sqrt{\mu_{r}}+\sqrt{\varepsilon_{r}}}\right|}^{2}$$

The reflectance of MMA is represented by Equation (5), which shows how impedance matching and metamaterial substantially impact it.

### 3.1. Absorbance Comparison with the Different Dielectric Layers and Metal

In this simulation, a dielectric layer and metal were used to simulate our proposed design (resonator and back-layer). The combination of tungsten and silicon dioxide shown in Figure 7a,b ensures the best result. On the other hand, as it is widely known that a high absorbance level is not always required for an application, several other applications using various dielectrics or metal layers have been covered in this article. In the first step, Figure 7a displays the results of replacing silicon dioxide with four other materials that serve as dielectrics. They are gallium arsenide (GaAs), silicon nitride, quartz, and silicon. It is clear from this comparison that silicon dioxide has a much higher average absorbance rate compared to other materials. The various refractive indices of such materials are the main cause of this sort of difference in the results. It is widely known that a lower refractive index also results in more excellent absorbance and broader bandwidth. The refractive index of Si_3_N_4_, GaAs, Silicon, Quartz, and SiO_2_ is 2, 3.9, 3.88, 1.55, and 1.50, correspondingly. On the other hand, Figure 7b depicts a simulation of the unit cell using a variety of metals, including aluminum, copper, platinum, silver, and tungsten. Tungsten is suitable for the proposed cell due to its complete optical area and excellent impedance match. Due to the incompatibility in impedance with this structure, aluminum, platinum, and copper exhibited poor absorbance. Here, silver and tungsten denote a high absorbance level of more than 80% between 450 and 650 nm.

### 3.2. Geometric Parameter Sweep

A parametric analysis is required to understand the proposed MA’s absorbance properties. The sweeping parameter plays an important role in determining the best possible design and utilization absorbance. The parameter of H1 represents the thickness of tungsten at the bottom plane. Figure 8a shows that the absorbance property is studied in the wavelength range of 30–110 nm for H_1_. It offers a unique absorbance curve for greater thicknesses with H_1_ = 100 nm. To maximize absorbance and prevent transmission, the MMA design used H_1_ = 110 nm, and absorbance levels are also impacted by the dielectric thickness (SiO_2_). Figure 8b shows the absorbance at various dielectric thicknesses for the parameter of H_2_. When we used H_2_ = 30 nm, the lower optical wavelength is absorbed more than the upper visible wavelength range. Moreover, absorbance values in the upper visible wavelength range are greater for thicker materials than those at the lower optical wavelengths. The highest average absorbance takes place at H_2_ = 60 nm. The MMA patch thickness is also studied for H_3_. There is a constant amount of absorbance for different H3 values, but there is a slight distortion in the upper optical range. The value of H_3_ = 15 nm has been determined as the highest absorbance level. A consistent absorbance bandwidth is achieved, and very few deformations are observed in the higher wavelength, as shown in Figure 7c. The resonator of the MMA was also studied. The value of L_2_ = 205, W_2_ = 480, W_3_ = 180.03, and W_4_ = 50 nm has been determined as the highest absorbance level. The circular ring resonator (CRR) considers R_1_ = 140 and R_2_ = 160 nm, respectively.

### 3.3. Co-polarization and Cross-Polarization with Polarization Conversion Ratio (PCR)

The proposed MMA’s PCR value was determined to verify that the proposed MMA is not a polarization converter but an absorber. As shown in Figure 9a, we demonstrated both co-polarization and cross-polarization components with the help of Equation (6). It is clear from Figure 9 that the cross-polarization component is nearly 0 on the linear magnitude scale. Thus, it was ensured that no waves were converted by this design.
| *S*_11_(*ω*) |^2^ = | *S_TE_*_, *TE*_ (*ω*) |^2^ + | *S_TE_*_, *TM*_ (*ω*) |^2^ = *R_xx_*^2^ + *R_xy_*^2^(6)

The PCR value is considered with the help of Equations (7) and (8), one for y-polarization and the other for x-polarization, respectively. Figure 9b depicts the PCR value of both polarized waves, showing that the suggested design is an absorber but not a polarization converter. The figure clearly shows that the PCR value computed close to 0 due to very little cross-polarization.
(7)$$Y\;\mathrm{to}\;x\;\mathrm{polarized}\;\mathrm{wave},\;{\mathrm{PCR}}_{y}=\frac{R_{xy}{}^{2}}{\left(R_{yy}{}^{2}+R_{xy}{}^{2}\right)}$$
(8)$$X\;\mathrm{to}\;y\;\mathrm{polarized}\;\mathrm{wave},\;{\mathrm{PCR}}_{x}=\frac{R_{yx}{}^{2}}{\left(R_{xx}{}^{2}+R_{yx}{}^{2}\right)}$$

### 3.4. Polarization Independence and Oblique Incident Angle Stability

Its polarization insensitivity was evaluated to confirm the suggested MMA’s absorbance efficiency. Figure 10a,b demonstrate the absorbance properties of various incident angles (ϕ) for TM and TE modes. In the TE mode, the *z*-axis determines the propagation direction of the wave, and its location along the *z*-axis defines the magnetic field vector (Hz). The *x* and *y*-axes represent the electric (E_x_) and magnetic field vectors (H_y_). Moreover, the TM mode’s electric field vector (E_z_) is long in the direction of wave propagation. The electric (E_y_) field vector and the magnetic (H_x_) field vector corresponding along the x and y axes. The proposed MMA obtained an exceptional absorbance characteristic for polarization incidence angles (ϕ) up to 90° due to its axial and rotational symmetry. The EM wave hits the MMA at an oblique angle, while all previous results were at a standard angle (θ = 0°). As a result, the explanation of oblique incidents absorbance behavior is also essential. The MMA structure is designed to have an oblique incidence angle (θ) between the z-axis and the direction of wave propagation. Figure 11a,b illustrate the absorbance curves of both modes of TE and TM for theta (θ) up to 70°, where above 70% absorbance is achieved.

### 3.5. Bendable Property of MMA

The MMA sheet may be bent during installation for external forces. Absorbance in both TE modes and TM modes has been studied for the banding effect. An absorbance curve with bending angles θ = 0° to 18° with three degrees distances is shown in Figure 12. The absorbance increases at lower wavelengths and decreases at higher wavelengths in TE mode. However, the TM mode demonstrates the same pattern of absorbance characteristics. Due to the unique absorbance property, the proposed absorber has a band-able property in the TE and TM modes, which is achieved at different banding angles.

### 3.6. Electric Field, Magnetic Field, and Surface Current Analysis

Figure 13 and Figure 14 depict the e-field (V/m) and h-field (A/m) distributions of a proposed MMA at a normal incidence angle of 0°. This electromagnetic activity can also be used to determine the absorbance mechanism. The electromagnetic properties of the absorber are increased and disturbed in a new area. Moreover, electromagnetic behavior is greatly impacted by the metamaterial property. The e-field and h-field characteristics of the wavelengths at 450 nm, 557 nm, and 650 nm are discussed. The correlation between electromagnetic performance and metamaterial characteristics can be shown in Equations (9) and (10).
(9)$$B=\mu_{eff}\;\mu_{0}H$$
(10)$$D=\varepsilon_{eff}\;\varepsilon_{0}E$$
where $\varepsilon_{eff}$ = effective permittivity, $\mu_{0}$ = permeability of free space, $\varepsilon_{0}$ = permittivity of free space, $\mu_{eff}$ = effective permeability, E = the electric field intensity and H = magnetic field intensity, B = the magnetic flux density, and D = the electric flux density. The highest e-field (4.24 × 10^7^ V/m) is attained in the patch’s bounding area, whereas the h-field maximum intensity (1.28 × 10^5^ A/m) appeared in the patch’s metal region. The non-metallic portion of the dielectric medium has a strong e-field, while the SRR’s left and right arms have a strong magnetic field. The e-field intensity is lowered at 450nm, as shown in the absorber. This results in a lower 90% absorbance at 450 nm and 650 nm, respectively, instead of a higher 99.99% absorbance at 557 nm. On the other hand, Figure 15 depict the surface current (A/m) distributions of a proposed MMA at a normal incidence angle of 0°. It would seem that the surface current transfer at 450 nm was followed by a surface current that was much stronger when it reached 557 nm independently. Figure 15c demonstrated that the surface current dispersion was extremely substantial at 557 nm. In each of the three stages, the rear layer exhibited lower energization levels than the front layer.

### 3.7. Aspect Ratio and Viability about Practical Fabrication

The aspect ratio describes the relationship between any material structure’s width and height. When the aspect ratio is high, the metal layer’s pattern, thickness, and shapes stand a better chance of being preserved in the construction. A substantial ground layer of tungsten contributes to the formation of a matched front-to-back metal layer. In addition, the absorber design is probably Microforming-friendly. Microforming, sometimes referred to as high aspect ratio micromachining (HARM), is a technique where metal elements are added into moulds made from extremely thick, patterned layers of photoresist. The LIGA microforming process allows for the patterning of X-ray photoresist PMMA in thick layers without the risk of diffraction, which enables the creation of devices with very high aspect ratios [32]. On the other hand, nanoimprint lithography and soft lithography are two other nanofabrication methods. Microcontact printing, nanoimprint lithography, and replica moulding are all examples of soft lithography techniques [33]. In the proposed structure, silica dioxide serves as the substrate material; thus, the first step is to construct a master layer on top of the substrate. Then, the elastomeric component is applied in the creation of a soft mould for the purpose of imprinting. All of these different manufacturing processes verify the fact that the proposed nano structure can be fabricated.

### 3.8. CST and HFSS Validation

The proposed structure was designed, and its performance was studied using computer simulation software (CST). This research verified the suggested MMA unit cell performance using the high-frequency simulation software (HFSS). Figure 16 shows the acquired data from several simulation programs that show the results of comparing CST and HFSS. This research found that both HFSS and CST software show transverse electric (TE) and transverse magnetic (TM) modes of propagation. The suggested MMA has a maximum absorbance of 99.99% at 557 nm and an average of 91.82% from 450 to 650 nm at TE and TM mode when measured using CST. On the other hand, a wavelength range of 450 to 650 nm with an average absorbance of 96.45% in TE and TM mode when the suggested MMA unit cell performance was verified using the HFSS software. Finally, the plot shows that both CST and HFSS simulation data are acquired.

## 4. Comparative Study

The results of the performance comparison between our proposed research and previously published research are presented in Table 2, where the size of the MMA, material, absorbance, polarization, incident angle stability, and operation range are compared. The findings indicate that the proposed MMA we designed is more compact than the study provided as a reference. In comparison to the previous research, the absorbance of the MMA that has been proposed is satisfactory. The proposed SiO_2_ and W-based MMA offers 70° stability for absorbance levels greater than 70%, while operating over a more extensive working bandwidth. Moreover, a peak absorbance rate of 99.99% is also attained. However, the proposed design is appropriate for absolute invisible layers, substantial absorbance, wide-angle stability, magnetic resonance imaging (MRI), color images, thermal imaging, solar cells, etc.

## 5. Conclusions

This paper proposed a three-layer MMA for visible optical wavelength application. The predicted design evaluation and geometrical characteristics have been studied to achieve near-unity absorbance. The PCR value demonstrates that the proposed design absorbs relatively more than a polarization converter. TE mode and TM mode absorbance calculations are shown simultaneously, exhibiting equal absorbance and increasing the acceptance of the suggested MMA. The proposed MMA demonstrated an average absorbance of 91.82% from a wavelength of 450 nm to 650 nm, where the maximum absorptivity of 99.99% and the smallest size of 800 nm × 800 nm × 175 nm compared with the reference paper. There is no polarization sensitivity in either the TE or TM modes or the width oblique incident angle stability of 60°. The MMA is also demonstrated by 18° bending effects in both methods. Comparing the proposed MMA’s unique features with those of existing MMAs shows the advantages of the new MMA over the old. CST and HFSS simulation software have done the proposed MMA validation, the results of which match each other. Finally, the proposed MMA has an attractive solution for visible optical wavelength applications. Moreover, the study may also serve as a guide for future ATW research.

## Figures and Tables

**Figure 1 nanomaterials-12-04253-f001:**
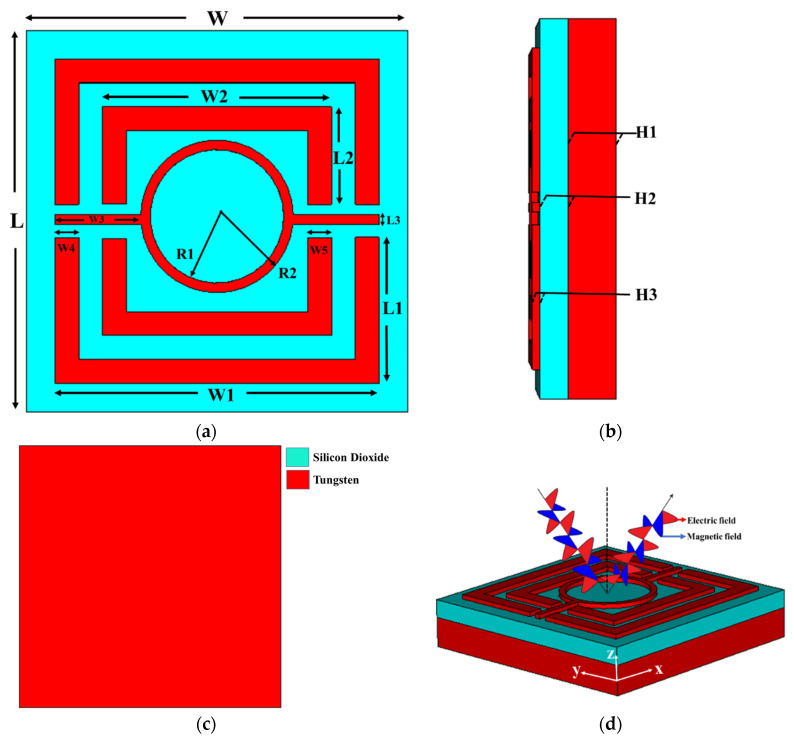
The design of the unit cell for visible optical wavelength applications (**a**) front view (**b**) side view (**c**) back view (**d**) perspective view.

**Figure 2 nanomaterials-12-04253-f002:**
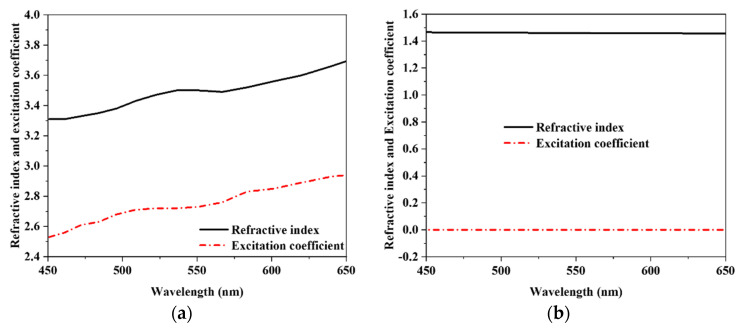
(**a**) Tungsten’s and (**b**) Silicon dioxide’s excitation coefficient and refractive index in the visible optical range.

**Figure 3 nanomaterials-12-04253-f003:**
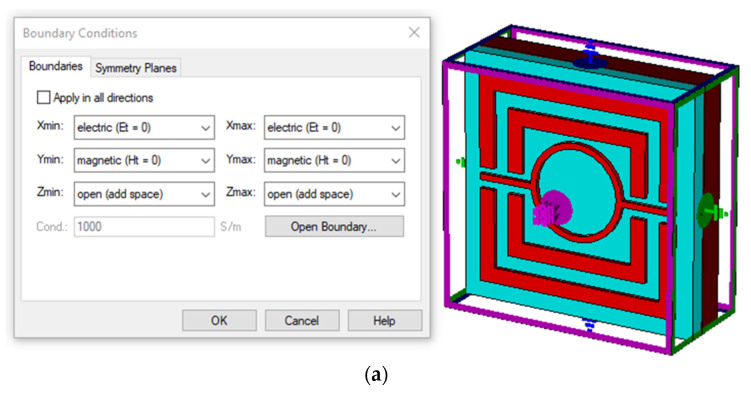
Setup of boundary conditions (**a**) PEC and PMC for TEM mode (**b**) Array configuration for TM and TE mode (**c**) The characteristics of the tetrahedral mesh.

**Figure 4 nanomaterials-12-04253-f004:**
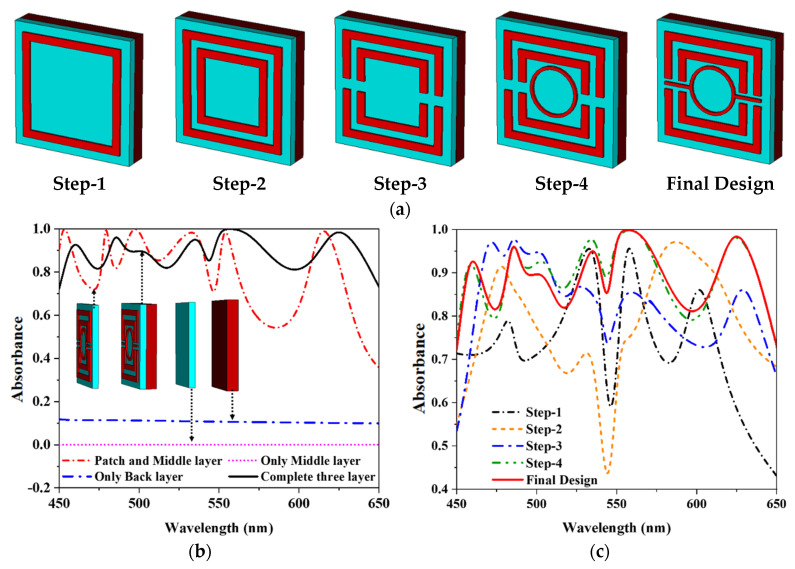
The absorbance of unit cell (**a**) design step (**b**) different layers and (**c**) design evolution.

**Figure 5 nanomaterials-12-04253-f005:**
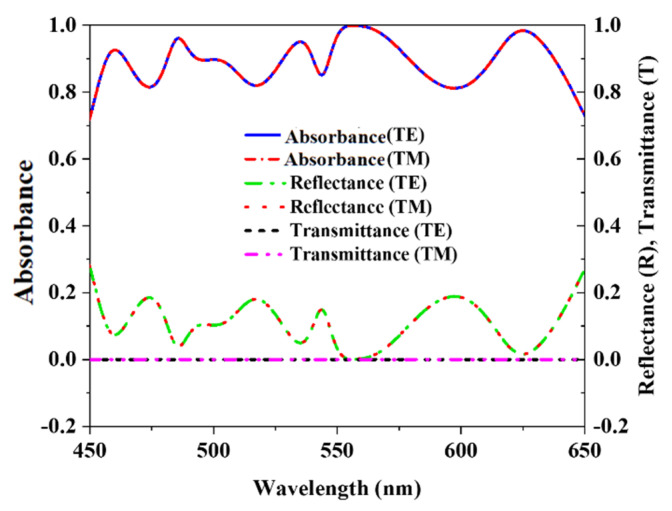
Reflectance, Transmission, and Absorbance plot.

**Figure 6 nanomaterials-12-04253-f006:**
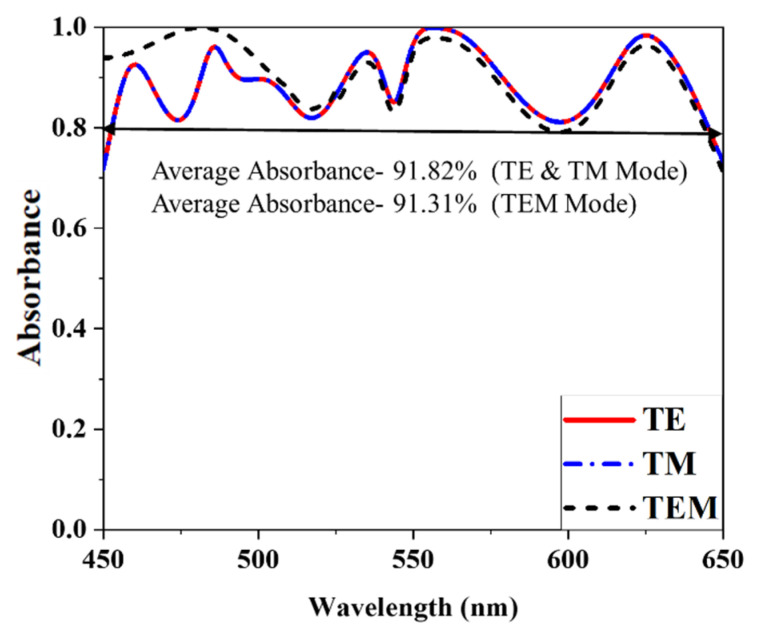
Absorbance characteristics comparison between TE, TM, and TEM modes.

**Figure 7 nanomaterials-12-04253-f007:**
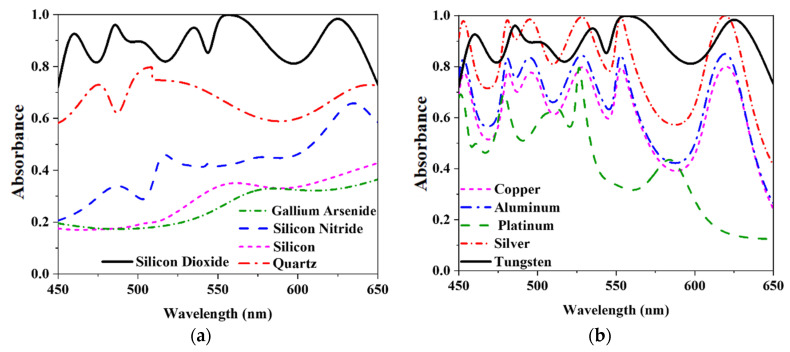
The absorbance compared with (**a**) several dielectric layers and (**b**) various types of metals.

**Figure 8 nanomaterials-12-04253-f008:**
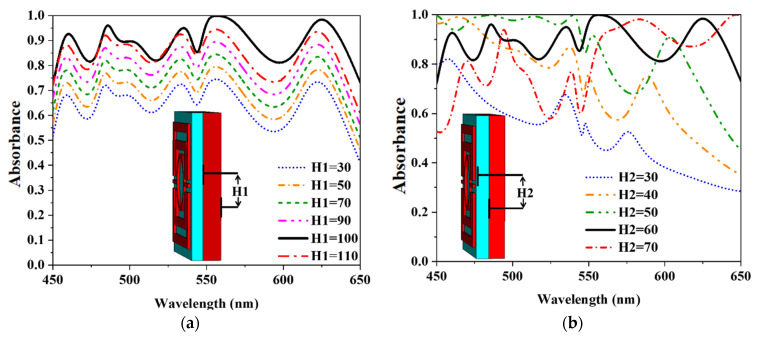
Parametric analysis of (**a**) the bottom tungsten layer thickness, (**b**) SiO_2_ dielectric layer thickness, (**c**) resonating patch of tungsten layer thickness, (**d**) resonating patch’s wide (**e**) effect of the circle leg (**f**) effect of the circle radius.

**Figure 9 nanomaterials-12-04253-f009:**
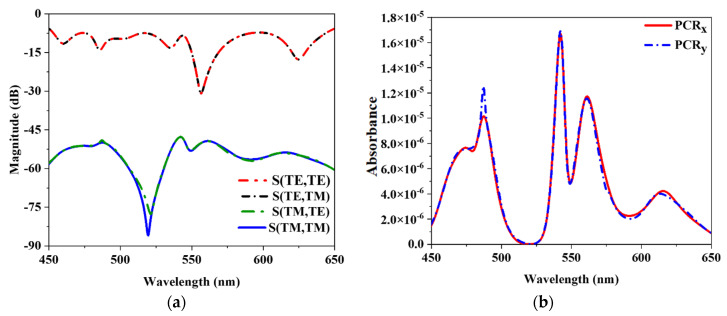
(**a**) Co-polar and Cross-polar component (**b**) The PCR value of x-polar and y-polar of the incident wave.

**Figure 10 nanomaterials-12-04253-f010:**
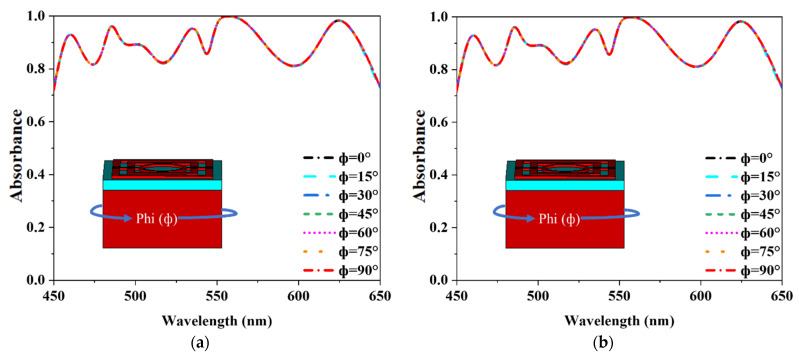
Illustration of the polarization independency of the MA (**a**) TE mode and (**b**) TM mode.

**Figure 11 nanomaterials-12-04253-f011:**
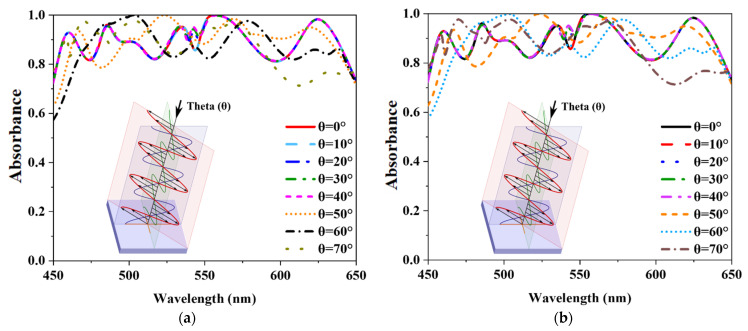
Angular stability characteristics of the absorber (**a**) TE mode (**b**) TM mode.

**Figure 12 nanomaterials-12-04253-f012:**
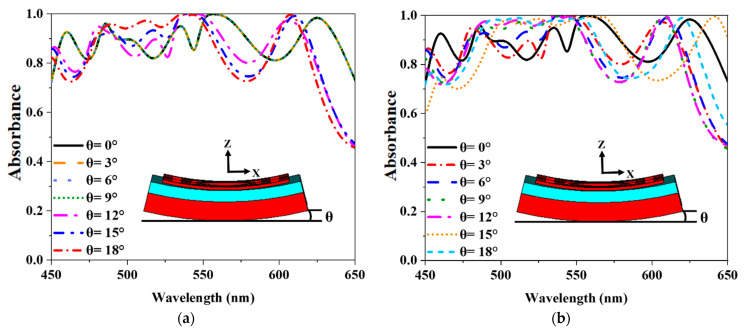
The proposed MMA bending effect (**a**) TE mode (**b**) TM mode.

**Figure 13 nanomaterials-12-04253-f013:**
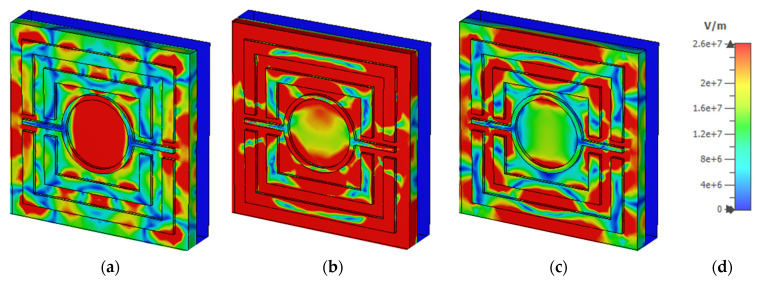
E-field distribution at (**a**) 450 nm (**b**) 557 nm (**c**) 650 nm (**d**) V/m scale.

**Figure 14 nanomaterials-12-04253-f014:**
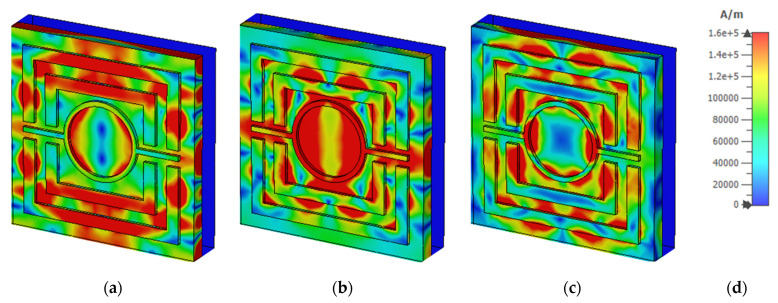
H-field distribution at (**a**) 450 nm (**b**) 557 nm (**c**) 650 nm (**d**) A/m scale.

**Figure 15 nanomaterials-12-04253-f015:**
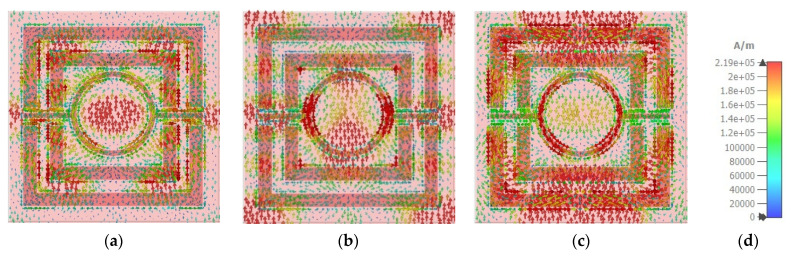
Surface current distribution at (**a**) 450 nm (**b**) 557 nm (**c**) 650 nm (**d**) A/m scale.

**Figure 16 nanomaterials-12-04253-f016:**
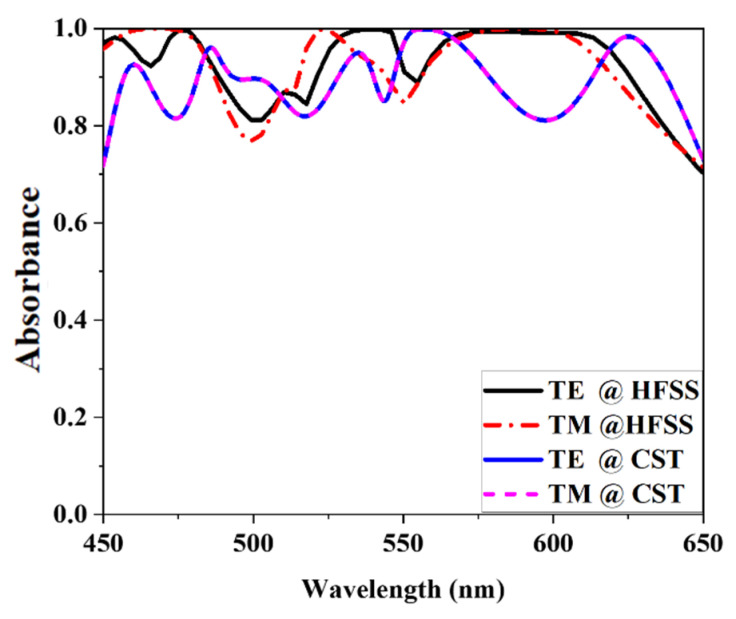
The proposed MMA absorbance comparison with different simulation software.

**Table 1 nanomaterials-12-04253-t001:** MMA unit cell’s structural parameters.

Parameters	L	L_1_	L_2_	L_3_	W	W_1_	W_2_	W_3_	W_4_	W_5_	H_1_	H_2_	H_3_	R_1_	R_2_
**Value (nm)**	800	306	205	20	800	680	480	180.03	50	50	100	60	15	140	160

**Table 2 nanomaterials-12-04253-t002:** Comparison table.

Ref.	Material	Size (nm)	Absorbance (Above)	Peak Absorbance	Operation Range (nm)	BW (nm)	Polarization Independence	Incident Angle Stability
[23]	Tungsten, SiO_2_	1000 × 1000 × 225	91.24%	99.99%	350–700	308	Yes	60°
[25]	Au, Si	500 × 500 × 600	80%	98.5%	474–784	310	Yes	60°
[26]	Ni, Si	250 × 250 × 355	90%	99%	400–700	300	Yes	60°
[34]	Tungsten, SiO_2_	133.3 × 133.3 × 99	90%	85.2%	428–1070	642	Yes	30°
[35]	Tungsten, SiO_2_	200 × 200 × 206.59	80%	98%	350–1200	850	Yes	45°
[36]	Ag, SiO_2_	120 × 120 × 60	70%	92%	430–750	320	Yes	40°
[37]	Tungsten, Quartz	1000 × 1000 × 120	80%	99.99%	450–600	150	Yes	60°
**Proposed**	**Tungsten, SiO_2_**	**800 × 800 × 175**	**91.82%**	**99.99%**	**450–650**	**200**	**Yes**	**70°**

## Data Availability

Data already plotted in the manuscript in different figures.

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
