# Peer review of "Design and Parametric Analysis of a Wide-Angle and Polarization Insensitive Ultra-Broadband Metamaterial Absorber for Visible Optical Wavelength Applications"

_nanomaterials, 2022, doi:10.3390/nano12234253_

Round 1

Reviewer 1 Report (Previous Reviewer 1)

Accept in present form.

Author Response

Thank you very much for your positive comments. 

Reviewer 2 Report (Previous Reviewer 2)

performance of metasurface structure is modelled and verified by different modelling techniques. methods required for fabrication are discussed. 

obviously,  experimental verification would bring strong advance for this project and is fully encouraged.  

one issue should be solved. what is plotted is "Absorbance" which is a measure of physical phenomenon of "Absorption". absorption is not quantified, absorbance is quantification of absorption. this would help with presentation and discussion of results. 

Author Response

Reviewer#2, Concern #1: Comments and Suggestions for Authors: performance of meta surface structure is modelled and verified by different modelling techniques. methods required for fabrication are discussed. Obviously, experimental verification would bring strong advances for this project and is fully encouraged. one issue should be solved. What is plotted is "Absorbance" which is a measure of physical phenomenon of "Absorption". absorption is not quantified; absorbance is quantification of absorption. This would help with presentation and discussion of results.

Author response: Thank you very much for your suggestions in our article. We measured how much our nano-unit cell absorbed for visible optical range. We corrected our all plots e.g., figure 4-12. As well as presentation and discussion of results in “Absorbance”. Author action: We updated the manuscript based on reviewer comments.

Reviewer 3 Report (Previous Reviewer 3)

The authors have substantially revised the manuscript and addressed methodology and analysis issues. I suggest that the work is published after minor English language editing.

Author Response

Reviewer#3, Concern # 1:

Comments and Suggestions for Authors:

The authors have substantially revised the manuscript and addressed methodology and analysis issues. I suggest that the work is published after minor English language editing.

Author response:  Thank you very much for your suggestion. Based on your comments we have checked and edited our article English in the revised manuscript which are marked in red color.

Reviewer 4 Report (Previous Reviewer 4)

Authors have extensively revised their manuscript. I think the paper is now ready to be published.

Author Response

Reviewer#4, Concern # 1:

Comments and Suggestions for Authors:

Authors have extensively revised their manuscript. I think the paper is now ready to be published.

Author response:  Thank you very much for your positive and constructive comments.

This manuscript is a resubmission of an earlier submission. The following is a list of the peer review reports and author responses from that submission.

Round 1

Reviewer 1 Report

In this study, the authors used a three-layer (metal dielectric metal) substrate material configuration to numerically analyze the compact split ring resonator (SRR) and the ring resonator based metamaterial absorber (MMA) for broadband visible light applications. The observable novelty of the proposed metamaterial is compact, and it achieves an average absorption of 97.82% for visible optical wavelength. The MMA has diverse applications such as wide-angle stability, substantial absorption,  magnetic resonance imaging (MRI), absolute invisible layers, thermal imaging, and color image applications. I believe that publication of the manuscript may be considered only after the following issues have been resolved.

1.      The author mentioned that the three-layer structure of metal dielectric metal is adopted in this work, but in the model of the article, the structure is dielectric metal dielectric three-layer structure. I hope the author will check it carefully.

2.      In Figure 4, the author's explanation is too simple, and the author needs to describe it in more detail. In addition, the physical mechanism of the change of optical characteristics caused by the change of some structural parameters needs to be given by the author.

3.      The introduction can be improved. Some works on metamaterial absorber (MMA) and theirs related properties should be added such as Phys. Chem. Chem. Phys., 2022, 24, 8846 – 8853; Physical Chemistry Chemical Physics, 2022, 24, 4871 – 4880. As for formula 1-4, the following relevant references need to be mentioned by the author, such as: Plasmonics 2018, 13, 345–352; Plasmonics 2015, 10, 1537–1543.

4.      In Figure 8, how does the author change the angle in the software? And why does this angle change cause the optical performance to change?

Reviewer 2 Report

perfect absorbers are well developed. in the presented numerical study there  is no new knowledge presented. abstract should present new findings only.

fabrication, characterisation and numerical model would make sense for this type of study

it is already well demonstrated that interfaces are key for performance of metamaterials. the design and real structure can show remarkably different results. 

Reviewer 3 Report

The work investigates the broadband response of metamaterial absorbers. Compact split ring resonators and circular ring resonator based metamaterial absorber are considered, using a three-layer (metal-dielectric-metal) substrate material configuration. Numerical results have indicated substantially high absorption capabilities. Sensitivity studies have been as well performed to investigate the absorption properties at different incidence angles. The work is within the scope of the journal and overall of interest in the design of metamatetrial absorbers. I suggest that the work is published after a thorough language proof-reading, typo and grammatical error corrections, as well as after a more detailed description of the numerical modeling part (convergence and computational cost analysis is insufficient).

Reviewer 4 Report

In this paper, wide-Angle and polarization insensitive ultra-broadband
metamaterial absorber is investigated by using commercial software CST
studio.  I think the presentation is not good. This paper should be resubmitted
after rewriting.

1) I cannot exactly identify the structure shown in Fig. 1.
    Please show the width of inner rectangle patch and horizontal line patch connected to circle patch.
    What is H1, H2, and H3 ?  What is the bold black line in Fig. 1(b) ?
    Are these thickness of each layer ?

2) I think that the figure shown in Fig. 2(a) may be the screen shot of CST studio.
    They are not appropriate to use academic paper because these figures
    are not understandable by except for the software users.

3) Figure number is incorrect in Section 2.3. Figure 2 may be Figure 4.

4) I cannot understand why SiO2 (shown with red color) has absorption. 
     Because extinction coefficient is almost zero according to Fig. 2(b).
     Is the color correct in Fig. 1 ?

5) Equation number on 248th line is incorrect.

6) Please explain Eq.(2), (3), and (4) in more detail.
     How are these equation derived ?

7) In Fig. 6, are the structural parameters optimized for each material?
    Please show the structural parameters for each material.

8) It is important to mention about the concept of this structure.
     How is this structure considered ?

9) It is said that the average absorption is 97.82% from 450 to 650 nm.
    However, from Fig. 4(a), I cannot believe the average absorption is such high.
    If this value is incorrect, the absorption property is not better than that in [21].